# A3VLM: Actionable Articulation-Aware Vision Language Model

**Siyuan Huang**[*]
Shanghai Jiao Tong University
Shanghai AI Laboratory

**Haonan Chang**[*]
Rutgers University

**Yuhan Liu**
Rutgers University

**Yimeng Zhu**
Yuandao AI

**Hao Dong**
Peking University

**Abdeslam Boularias**[†]
Rutgers University
abdeslam.boularias@rutgers.edu

**Peng Gao**[†]
Shanghai AI Laboratory
gaopeng@pjlab.org.cn

**Hongsheng Li**
CUHK MMLab
Shanghai AI Laboratory
CPII under InnoHK

**Abstract:** Vision Language Models (VLMs) have received significant attention in recent years in the robotics community. VLMs are shown to be able to perform complex visual reasoning and scene understanding tasks, which makes them regarded as a potential universal solution for general robotics problems such as manipulation and navigation. However, previous VLMs for robotics such as RT-1 [1], RT-2 [2], and ManipLLM [3] have focused on directly learning *robot-centric* actions. Such approaches require collecting a significant amount of robot interaction data, which is extremely costly in the real world. Thus, we propose A3VLM, an *object-centric*, actionable, articulation-aware vision language model. A3VLM focuses on the articulation structure and action affordances of objects. Its representation is robot-agnostic and can be translated into robot actions using simple action primitives. Extensive experiments in both simulation benchmarks and real-world settings demonstrate the effectiveness and stability of A3VLM. We release our code and other materials at https://github.com/changhaonan/A3VLM.

**Keywords:** LLM, VLM, Manipulation, Articulation

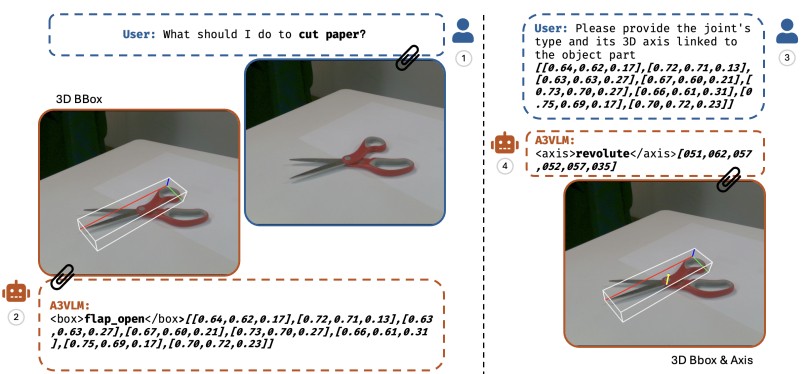

Figure 1: Sequential inference with prompts. To answer the first question, A3VLM identifies the corresponding action type and bounding box for the movable part of the scissors. Then, the second answer reveals the articulation structure of the scissor leg. A manipulation action can be performed based on the answer.

8th Conference on Robot Learning (CoRL 2024), Munich, Germany.

# 1   Introduction

The use of Large Language Models (LLMs) [4] for robotics applications has recently attracted significant attention. Combined with robot control APIs [5, 6], open-vocabulary scene graphs [7], or motion planners [8], LLMs have demonstrated an extraordinary ability to understand users' commands, reason about the environment, and select the correct action from skill pools. In these works, LLMs usually rely on external vision tools such as an open-vocabulary detector to convert scene information into text. This straightforward combination, however, prevents LLMs from acquiring further scene understanding and certain important details about the environment. Subsequently, researchers have begun to directly train Vision Language Models (VLMs) for manipulation and navigation tasks. Compared to methods that combine LLMs with external vision tools, native VLMs can capture detailed visual data and perform complex visual reasoning, making it possible to be an all-in-one solution for solving general manipulation tasks. However, the training of VLMs is extremely data-hungry, and the collection of image-text pair data for robotics VLMs can be a significant problem. Furthermore, the output of VLMs is purely text-based, which is fundamentally different from the standard robot action representation as trajectories. Thus, robot VLMs need to define a tailored and precise action representation for a robotic system.

There have been several attempts to address the data gathering and the action representation problems. RT-1 [1] and RT-2 [2] directly represent the robot's end-effector pose using a discretized 6D pose and collect a large amount of image-action pair data using teleoperation. ManipLLM [3] simplifies this action representation by replacing the parallel gripper with a suction gripper, requiring only the computation of a contact point and the gripper's direction. Interaction data is then collected in simulation. Such large amounts of robot interaction data are expensive to collect in the real world.

To address this issue, we shift our focus from directly learning actions to learning an object-centeric representation that is independent of the robot's configuration and that can easily be translated into low-level manipulation actions. We propose the Actionable Articulation-Aware VLM (**A3VLM**), with a representation that describes the object's articulation structure and action affordance simultaneously. Compared to previous robot-centric action representations [2, 3], A3VLM's representation is object-centric, which makes it possible to learn actionable models of objects without collecting expensive robot interaction data, and the same learned object models can be used by various robots.

Given a single RGB image of an unknown object and a language task description, A3VLM locates an actionable part of the object and provides necessary articulation information for a manipulation action. Results on the PartNet-Mobility simulation benchmark show that our proposed A3VLM outperforms previous related models by a large margin. Extensive real-world experiments also demonstrate A3VLM's excellent robustness and potential for real-world robot manipulation applications.

# 2   Related Work

**Manipulation of Articulated Objects.** An articulated object refers to an object composed of multiple rigid parts connected by movable joints, such as a drawer or dishwasher. Articulated object manipulation is an important topic in robotics. A common practice for articulated object manipulation is to determine its articulation structure first and then manipulate it with predefined action primitives. Some works [9, 10, 11] use part-based pose estimation methods to locate different articulation structures. Other methods, such as FlowBot3D [12], predict the per-point articulation movement of every point on the object's 3D point cloud. A subsequent technique, FlowBot++ [13], predicts for each point an articulation parameter instead of a movement. Although effective, these methods only perform articulation estimation. GaPartNet [14] shares similar insights with our work by combining articulation and affordance. Object links in GaPartNet are classified into nine different articulation prototypes, each with a different canonical pose and affordance. Link poses and affordances are detected from a point cloud. Inspired by GaPartNet, A3VLM also predicts the articulation of each part of the object, as well as its corresponding action prototype. A3VLM simplifies GaPartNet's nine prototypes to only two types, *prismatic* and *revolute*. Lastly, most existing articulated object detection and manipulation methods [12, 13, 14] are based on 3D point clouds. 3D point-cloud data in real-world environments can be noisy and inaccurate due to reflections and transparency, as

illustrated in Fig. 6. With the support of a strong VLM backbone, A3VLM is able to predict 3D articulation structures directly from a single RGB image, without any depth data.

**LLMs/VLMs for Manipulation.** Existing LLMs/VLMs for manipulation can be divided into three main categories. The first type, such as Code-as-Policies [15], Instruct2Act [6], SayCan [16], and others, uses LLMs/VLMs to generate high-level semantic action plans in pure text or code. The system then relies on external low-level action models or hand-crafted APIs to execute these plans. This approach heavily depends on the implementation of the low-level skills and APIs, and it is primarily limited to simple tasks such as pick-and-place. In contrast, methods like RT-1 [2], RT-2 [2], and ManipLLM [3] aim to generate robot actions directly, which enables them to handle more complex manipulation tasks, such as opening drawers and closing doors. However, action-based LLMs/VLMs typically require a substantial amount of robot-environment interaction data, which can be costly to collect in the real world. Additionally, since this data is often collected from specific types of robots, it cannot be directly used for different robots. The third main category involves using LLMs/VLMs to generate intermediate representations, such as cost maps (VoxPoser [17]), action constraints (MOKA [18]), or affordances (ManipVQA [19]), which are then translated into robot actions using simple action primitives or controls. Our A3VLM falls into this third category. Unlike previous methods, A3VLM focuses on the articulation structure of objects, which enables complex manipulation actions. To the best of our knowledge, A3VLM is the first VLM capable of accurately and consistently locating and understanding the articulation structure for robot manipulation.

# 3 Method

## 3.1 Proposed Articulation Representation

Unlike the robot-centric action representation in RT-1 [1], RT-2 [2] and ManipLLM [3], A3VLM uses an object-centric representation that focuses on the articulations and affordances of the movable parts within an object. Compared to a single articulation detection pipeline such as FlowBot++ [13], A3VLM can predict the affordance of each part of the object and locate the appropriate part to manipulate based on the desired task. GapartNet [14] has a similar representation. However, GapartNet uses nine different types of articulation prototypes, whereas we unify articulation structures into two basic types: prismatic articulation and revolute articulation.

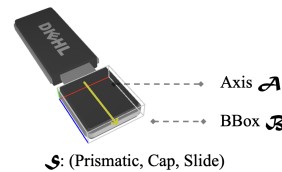

$\mathcal{S}$: (Revolute, Handle, Rotate)

$\mathcal{S}$: (Prismatic, Cap, Slide)

Figure 2: Articulation Representations in A3VLM. Top: Revolute handle (rotate). Bottom: Prismatic cap (slide).

Actionable parts, affordances, and articulation structures in A3VLM are represented as a triad: (Bounding box $\mathcal{B}$, Axis $\mathcal{A}$, Semantic label $\mathcal{S}$). Bounding box $\mathcal{B}$ locates an actionable part of interest in the given image. Axis $\mathcal{A}$ represents the articulation structure of the part. Semantic label $\mathcal{S}$ refers to the articulation type (prismatic or revolute), the link name and the action type. Examples of this representation are shown in Fig. 2.

In practice, the 3D bounding box $\mathcal{B}$ is represented by its eight vertices $\{(x_i, y_i, z_i)\}_{i=1,...,8}$. Here, $(x_i, y_i)$ is the 2D projected position of vertex $i$ in the given image's plane, and $z_i$ is normalized to the range (0,1) using the maximum and minimum depth values. Axis $\mathcal{A}$ is represented using two edge points $\{(\alpha_i, \beta_i, \gamma_i)\}_{i=1,2}$ in the 3D space, and it is normalized using the same method used for the bounding box $\mathcal{B}$.

## 3.2 Instruction-following Dataset Construction

| Capabilities | Tasks | Examples of Task Templates | Num. |
|---|---|---|---|
| Partial Object Understand. | Detection | User: Detect all manipulable object parts and provide their 3D bounding boxes. 
 A3VLM: There is one manipulable object parts with their 3d bounding boxes: object name and its **BBox** $\mathcal{B}$. | 43K |
| Partial Object Understand. | REC-Link | User: Please provide the 3D bounding box of the region this sentence describes: lid. 
 A3VLM: **BBox** $\mathcal{B}$. | 178K |
| Articulation Understand. | REG-Joint | User: Please provide the joint's type and its 3D axis linked to the object part: **BBox** $\mathcal{B}$ or **Link Name** $\mathcal{S}$. 
 A3VLM: **Joint type** $\mathcal{S}$ and its **Axis** $\mathcal{A}$. | 18K |
| Action Grounding | REC-Action | User: Please execute the task described with 3D rotated bounding box representations by the following instruction: Open the storage. 
 A3VLM: **Action type** $\mathcal{S}$ and targeted object's **BBox** $\mathcal{B}$. | 15K |

Table 1: Overview of the instruction-following dataset, including task templates, associated capabilities, examples, and sample counts. From top to bottom, 4 different tasks are presented: Detection, REC-Link, REG-Joint and REC-Action.

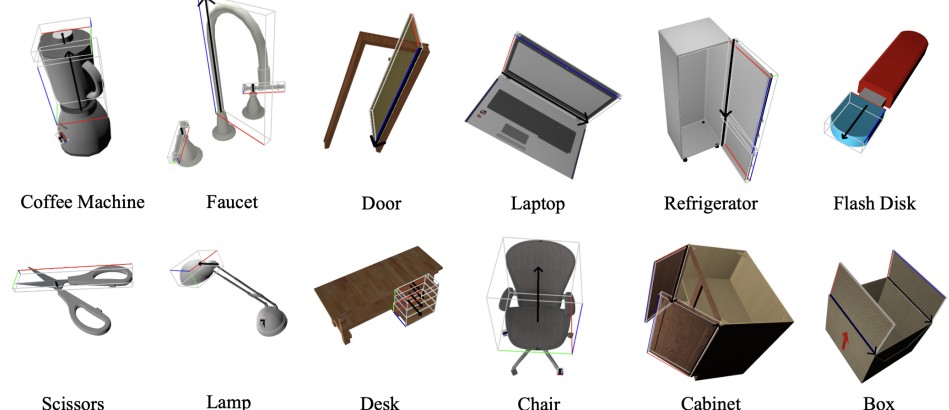

| Coffee Machine | Faucet | Door | Laptop | Refrigerator | Flash Disk |
| --- | --- | --- | --- | --- | --- |
| Scissors | Lamp | Desk | Chair | Cabinet | Box |

Figure 3: Annotations used for training **A3VLM** on the PartNet-Mobility [20] dataset. The white box illustrates the Bounding box $\mathcal{B}$ for articulation structure, while the black axis represents the Axis $\mathcal{A}$. This figure shows annotations from 10 different categories.

As training a VLM requires colossal resources in terms of data and computation, we do not train A3VLM from scratch. Instead, we fine-tune an established VLM. To fine-tune a VLM, we need to build an instruction-following dataset, where the input is an image and a text prompt, and the answer should be structured text. In this section, we will discuss how to construct this instruction-following dataset. As we mentioned in Sec. 3.1, we use a triad $(\mathcal{B}, \mathcal{A}, \mathcal{S})$ to describe the location, articulation structure, and action affordance of a movable part. In practice, we do not ask the VLM to generate everything in one inference step but separate the tasks into four different types of sub-tasks. This separation allows the VLM to focus on one concept during each inference.

**Raw A3 Annotation Generation.** The first step in generating an instruction-following dataset is to create object-level raw annotations. We use PartNet-Mobility [20], which provides more than 2,000 different articulated objects across 46 categories in URDF format. First, we render these objects into RGB images using PyRender[1], incorporating random camera positions, lighting, and joint values to generate 40 different images for each object. For each image, we use ControlNet [21] to generate augmented images for data augmentation (see Sec. 3.3 for more details on data augmentation).

Within each image, we provide an annotation $(\mathcal{B}, \mathcal{A}, \mathcal{S})$ for each visible and movable link. We categorize all links into prismatic and revolute types. For revolute links, axis $\mathcal{A}$ is the rotation axis provided in the URDF. For prismatic links, we use the prismatic direction provided in the URDF as the axis direction, ensuring that axis $\mathcal{A}$ passes through the 3D center of the link. After determining axis $\mathcal{A}$, we project the link points along $\mathcal{A}$ and compute a minimal 2D bounding box for the projected shape. We use the longer edge of this bounding box as the x-axis, the shorter edge as the y-axis, and axis $\mathcal{A}$ as the z-axis for the bounding box $\mathcal{B}$. The center of the bounding box $\mathcal{B}$ is the 3D center of the link. The width, height, and length of the bounding box $\mathcal{B}$ are computed based on the distance between the furthest points of the link and the center. Semantic information $S$ stores the articulation type, name, and affordable action of the link. Fig. 3 shows multiple annotation examples from different categories in the PartNet-Mobility dataset.

Noticeably, the affordable actions of the links in the PartNet-Mobility dataset are not provided. We therefore select the actions from a robotic skill library as defined in [22]. To ensure that the selected skills are compatible with the corresponding links of the objects, we use GPT-4 for skill selection. The prompt used during this process can be found in Appendix C.

**Sub-tasks Construction.** To fit into the established VLM training pipeline, we follow the widely used VLM task templates: Referring Expression Comprehension (REC) and Referring Expression Generation (REG). In a nutshell, REC requires the VLM to provide a bounding box according to a text description, while REG asks the VLM to provide a description of a region within an image referred to by a bounding box. The definitions of REG and REC can be found in [23].

---

[1] https://github.com/mmatl/pyrender

Following these definitions, we construct four different sub-tasks: (1) Detection, (2) REC-Link, (3) REG-Joint, and (4) REC-Action. Each sub-task consists of an image, a text question, and a text answer. Examples of each sub-task question can be found in Tab. 1. The detection task requires the VLM to locate all manipulable parts within an image by outputting a bounding box $\mathcal{B}$ for each part. Since this task generates more than one bounding box, it does not follow the traditional definition of REC tasks. The REC-Link task involves locating a part/link based on a description. The REG-Joint task asks the VLM to provide the joint axis $\mathcal{A}$ and the joint type $\mathcal{S}$ for a part specified by a bounding box $\mathcal{B}$. REC-Action requires the VLM to locate a movable part by providing a bounding box $\mathcal{B}$ and corresponding action type $\mathcal{S}$ according to an action task, such as "Open the storage."

### 3.3 Data Augmentation Strategy

A limitation of the original PartNet-Mobility dataset is the absence of texture details. As depicted in Figure 3, most objects are rendered in plain gray, which does not reflect real-world conditions. To address the simulation-to-reality (Sim2Real) gap, we employed ControlNet [21] to generate more realistic images, using depth maps as the primary control signal due to their ability to convey both geometric and semantic information. For objects with minimal depth variance, we utilized semantic segmentation as the control signal. To enhance the diversity of the generated images using Stable Diffusion, we employed ChatGPT to generate a broader range of detailed descriptions, enriching the contextual input necessary for producing varied visual outputs. Specific prompts used with ChatGPT and examples of the augmented data are detailed in Appendix G.

### 3.4 Model and Training

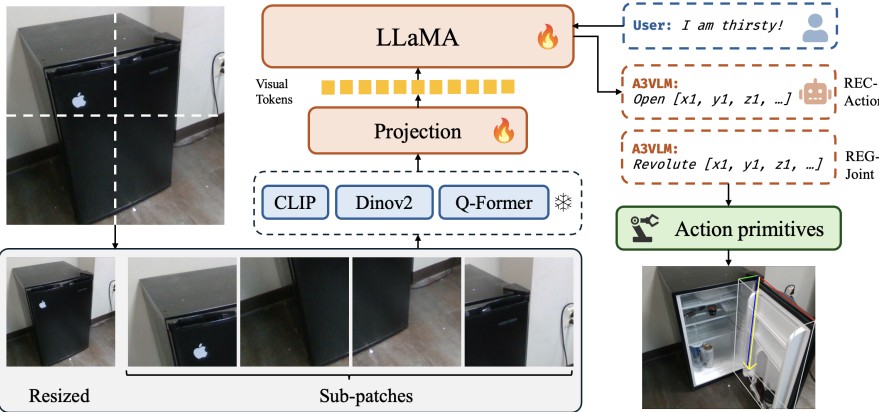

Figure 4: The A3VLM architecture. The input image is resized and divided into sub-patches before being processed by visual encoders to extract local and global visual features. These features are then concatenated and aligned with language tokens through projection layers. The fire emoji indicates that the projection layer and LLaMA are fine-tuned, while the snow emoji signifies that the visual encoders remain frozen.

**Model Architecture.** A3VLM is developed based on the SPHINX-X [24] with LLaMA2 serving as the language backbone. We select this model as it is uniquely tailored to focus on the partial or regional details of target objects, necessitating fine-grained visual analysis. Our architecture is shown in Fig. 4. Following the SPHINX "any resolution" approach [25, 24], the input image is first partitioned into sub-images and then visual encoders are applied to extract visual features. Moreover, given the necessity for both global and local visual grounding ability in manipulation tasks, we integrate the visual encoder from CLIP [26], DINOv2 [27] to extract local semantic features, and Q-Former [28] for the global visual features summarization. Then, local and global features are channel-wise concatenated. The spatial alignment between visual tokens and language tokens is achieved with projection layers. Bounding box values are normalized to the range $(0, 1)$ and expressed with precision up to two decimal places.

**Fine-tuning Strategy.** As discussed in Section 3.2, our training paradigm follows the conventional Visual Question Answering (VQA) framework and encapsulates all information about articulations within a natural language framework. As a result, the training objective only employs the cross-entropy loss, which is a departure from previous works [3, 29]. To bridge the visual disparity between our specialized dataset and generic natural imagery, we employ a two-stage fine-tuning

strategy. Initially, the visual projection layers are fine-tuned using straightforward image caption tasks, utilizing a basic template such as "This is a [OBJ]" to generate naive captions. Then, we fine-tune the visual projection layers and LLM simultaneously on the instruction-following dataset.

### 3.5 Action Primitives

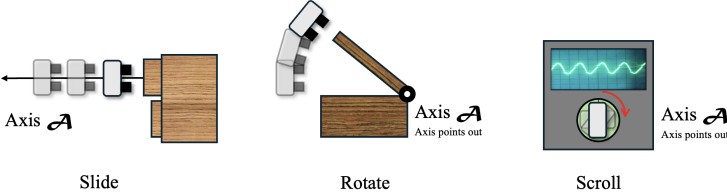

Figure 5: Action primitives for sliding, rotating, and scrolling. Prismatic joints use slide actions, revolute joints use rotate actions, and scroll actions are for semantically labeled targets like control knobs.

As previously mentioned, A3VLM utilizes an object-centric representation. To translate this into a robot movement, we need to define specific action primitives. A3VLM is designed for use with various types of robots; therefore, it is not optimized for any particular type of end-effector, such as a parallel or a suction gripper. An independent generic grasp-pose proposer is required to generate a list of grasp-pose candidates. During manipulation, we utilize the triad $(\mathcal{B}, \mathcal{A}, \mathcal{S})$ generated by A3VLM along with the grasp pose candidates.

We define three types of action primitives: *Rotate*, *Slide*, and *Scroll*. For a given link, if its corresponding joint type is prismatic, we select the slide action; if it is revolute, we choose the rotate action, unless the target link is semantically labeled as a bottle cap or scroll button, in which case we opt for the scroll action. If the action selected is "scroll", we ensure that the grasp pose overlaps with the rotation axis $\mathcal{A}$. Otherwise, we randomly select a grasp pose within the bounding box $\mathcal{B}$ to serve as the contact point $\mathcal{C}$. We then generate a trajectory using $\mathcal{C}$ and $\mathcal{A}$ for each action type, as illustrated in Fig 5. These trajectories constitute our generated actions.

## 4 Experiment

### 4.1 Implementation Details

We fine-tuned the A3VLM model using the SPHINX framework [24] on eight NVIDIA A100 (80 GB) GPUs. The fine-tuning was completed in three epochs, which took approximately 24 hours. The visual encoders were kept frozen throughout the fine-tuning phase to maintain the integrity of the pre-trained features. We utilized the SPHINX-1K model, sourced directly from the official repository, as our pre-trained base. Training was conducted with a batch size of 4 and a learning rate set to $2 \times 10^{-5}$. To effectively manage the computational load and enhance the training dynamics, we employed a gradient accumulation strategy with a factor of 4.

### 4.2 Qualitative Evaluation

To evaluate the action capabilities of A3VLM, we modified upon the settings of ManipLLM [3]. This benchmark utilizes Sapien [20] as the simulator and PartNet-Mobility [30] objects as the target objects. Object's joint values will be initialized to the middle value. We use a flying Franka Panda Robot's gripper with suction ability as the end-effector.

**Evaluation Metric**. The goal of this benchmark is to evaluate the model's ability to interact with the environment. In each task, we will load an articulated object from PartNet-Mobility dataset. Then, we will drive the end-effector to interact with the articulated object. The base of the object is fixed, so the robot needs to locate a movable part and understand the correct direction to drive the part. Regarding the definition of successful task, we follow the definition in ManipLLM, where we measure the movement of a part in the articulated objects as $d$. If, after manipulation, the part's movement $d$ exceeds a threshold $\sigma$, we define it as a success. We use the same threshold as ManipLLM, where $\sigma = 0.01^2$.

---

[2]ManipLLM uses two thresholds $\sigma = 0.01$ and $\sigma = 0.1$. But their main evaluation is performed under $\sigma = 0.01$.

**Baselines**. We compare the robot manipulation success rates across 20 training and 10 testing categories, evaluating our method against five baselines: (1).**Where2Act** [31]: This method processes point-cloud input to score potential contact points, selecting the highest scorer. It predicts 100 orientations for the end-effector and chooses the top-scoring one, modified here to use a suction gripper. (2).**UMPNet** [32]: This approach manipulates at the predicted contact point, aligning the end-effector's orientation perpendicularly to the object's surface.(3).**Flowbot3D** [12]: It identifies motion direction in point clouds, selecting the point with the greatest flow magnitude as the interaction point, with the flow direction determining the end-effector's orientation. (4).**Implicit3D** [33]: This framework uses the Transporter network to identify keypoints on 3D articulated objects, guiding the end-effector's pose for future tasks. (5).**ManipLLM** [3]: The state-of-the-art method on this benchmark, it employs a vision-language model to predict the contact point and forward direction of a suction gripper using an RGBD image and language prompt.

**Action Primitives Details**. For each object, we first detect a list of action parts with corresponding bounding boxes $\mathcal{B}$, axes $\mathcal{A}$, joint types, and link names $\mathcal{S}$. We select a random action part from the list and use its bounding box $\mathcal{B}$ and axis $\mathcal{A}$ to generate two robot trajectories. For example, for a handle of a faucet, we will generate trajectories to rotate it clockwise and counter-clockwise. We will execute these trajectories in two attempts. The task is regarded as successful if it succeed in either try.

**Result.** Table. 2 shows the performance of A3VLM and five other baselines. From the table, it is evident that A3VLM outperforms all baselines by a large margin for most object categories. We attribute this improvement to two factors. The first factor is the accurate grounding of actionable parts and articulation structure, and the second is the introduction of action primitives. Action primitives enable A3VLM to perform various actions on different articulated objects. Further discussion on the benchmark results can be found in Appendix.

| Method | [1] | [2] | [3] | [4] | [5] | [6] | [7] | [8] | [9] | [10] | [11] | [12] | [13] | [14] | [15] | [16] |
|---|---|---|---|---|---|---|---|---|---|---|---|---|---|---|---|---|
| Where2Act | 0.26 | 0.36 | 0.19 | 0.27 | 0.23 | 0.11 | 0.15 | 0.47 | 0.14 | 0.24 | 0.13 | 0.12 | 0.56 | 0.68 | 0.07 | 0.40 |
| UMPNet | 0.46 | 0.43 | 0.15 | 0.28 | 0.54 | 0.32 | 0.28 | 0.56 | 0.44 | 0.40 | 0.10 | 0.23 | 0.18 | 0.54 | 0.20 | 0.42 |
| FlowBot3D | 0.67 | 0.55 | 0.20 | 0.32 | 0.27 | 0.31 | 0.61 | **0.68** | 0.15 | 0.28 | 0.36 | 0.18 | 0.21 | 0.70 | 0.18 | 0.26 |
| Implicit3D | 0.53 | 0.58 | 0.35 | 0.55 | 0.28 | 0.66 | 0.58 | 0.51 | 0.52 | 0.57 | 0.45 | 0.34 | 0.41 | 0.54 | 0.39 | 0.43 |
| ManipLLM | 0.68 | 0.64 | 0.36 | 0.77 | 0.43 | 0.62 | 0.65 | 0.61 | 0.65 | 0.52 | 0.53 | 0.40 | 0.64 | 0.71 | **0.60** | **0.64** |
| Ours | **0.90** | **0.82** | **0.94** | **0.90** | **0.49** | **0.70** | **0.87** | 0.35 | **0.86** | **0.79** | **1.00** | **0.70** | **0.83** | **0.97** | 0.34 | 0.40 |

| Method | [17] | [18] | [19] | [20] | AVG | [T1] | [T2] | [T3] | [T4] | [T5] | [T6] | [T7] | [T8] | [T9] | [T10] | AVG |
|---|---|---|---|---|---|---|---|---|---|---|---|---|---|---|---|---|
| Where2Act | 0.13 | 0.18 | 0.13 | 0.40 | 0.26 | 0.18 | 0.35 | 0.38 | 0.28 | 0.05 | 0.21 | 0.17 | 0.20 | 0.15 | 0.15 | 0.21 |
| UMPNet | 0.22 | 0.33 | 0.26 | 0.64 | 0.35 | 0.42 | 0.20 | 0.35 | 0.42 | 0.29 | 0.20 | 0.26 | 0.28 | 0.25 | 0.15 | 0.28 |
| FlowBot3D | 0.17 | 0.53 | 0.29 | 0.42 | 0.37 | 0.23 | 0.10 | 0.60 | 0.39 | 0.27 | 0.42 | 0.28 | 0.51 | 0.13 | 0.23 | 0.32 |
| Implicit3D | 0.27 | 0.65 | 0.20 | 0.33 | 0.46 | 0.45 | 0.17 | 0.80 | 0.53 | 0.15 | 0.69 | 0.41 | 0.31 | 0.30 | 0.31 | 0.41 |
| ManipLLM | 0.41 | **0.75** | 0.44 | 0.67 | 0.59 | 0.38 | 0.22 | **0.81** | 0.86 | 0.38 | **0.85** | 0.42 | **0.83** | 0.26 | 0.38 | 0.54 |
| Ours | **0.62** | 0.50 | **0.90** | **0.73** | **0.91** | **0.88** | **0.76** | 0.74 | **0.86** | **0.79** | 0.67 | **0.96** | 0.50 | **0.62** | **0.72** | **0.76** |

Table 2: Comparisons of our method against baseline methods on PartNet-Mobility. The first 20 categories are training categories and the following 10 categories are testing categories.

## 4.3 Real-World Application

**Real-World Inference Test.** To test the stability and accuracy of A3VLM in the real world, we perform an inference test for A3VLM on many objects across different categories. It turns out A3VLM is able to correctly detect the movable parts of the objects and accurately recognize the articulation structure of those parts. As shown in Fig. 6, A3VLM consistently generates correct inferences on 20 different real-world objects. It is worth-noting that A3VLM is able to correctly perform inference on objects with reflecting or transparent surface (e.g. microwave oven, pot and coke bottle), which are very challenging for point-cloud-based methods due to their inaccurate depth.

**Real-World Robot Manipulation.** To test the effectiveness of the manipulation ability of A3VLM, we selected five different objects from the aforementioned twenty tested categories. We used a Kuka robot equipped with a RealSense D415 depth camera and a Robotiq three-finger gripper. We followed the action primitive mentioned in Sec. 3.5. Five trials with different initial positions are performed for each object. Manip-

Table 3: Manipulation success rates on the real-world objects.

| Object | [pot] | [bottle] | [microwave] | [spray] | [box] |
|---|---|---|---|---|---|
| Success | 5/5 | 5/5 | 5/5 | 5/5 | 4/5 |

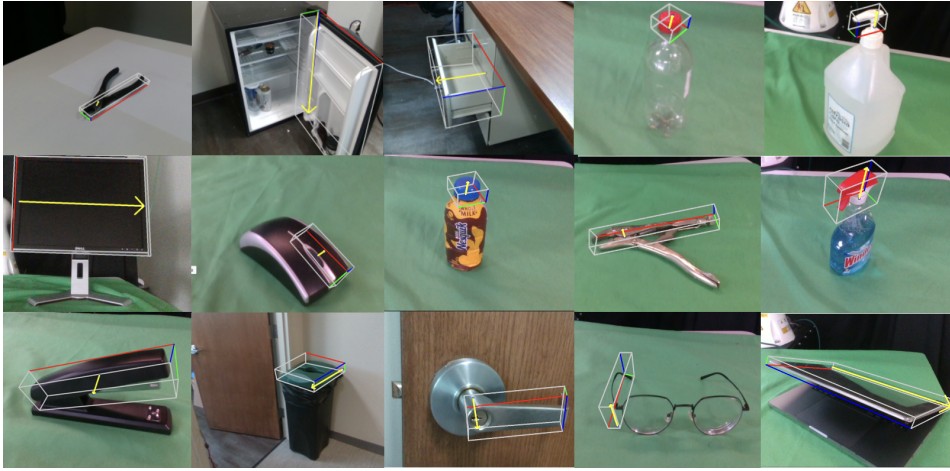

Figure 6: A3VLM predictions on various real-world objects using a single RGB image. A3VLM is able to perform correct inference on challenging real world objects, even for those with transparent or reflecting surfaces.

ulation is considered successful if the manipulated part has successfully moved more than 40% of its joint limit. Success rate results can be found in Tab. 3, and experiment start and end states are shown in Fig. 7. As shown in Tab. 3 and Fig. 7, A3VLM successfully manipulated the articulated objects in the real world with simple action primitives. Further implementation details on the robot experiments can be found in Appendix D.

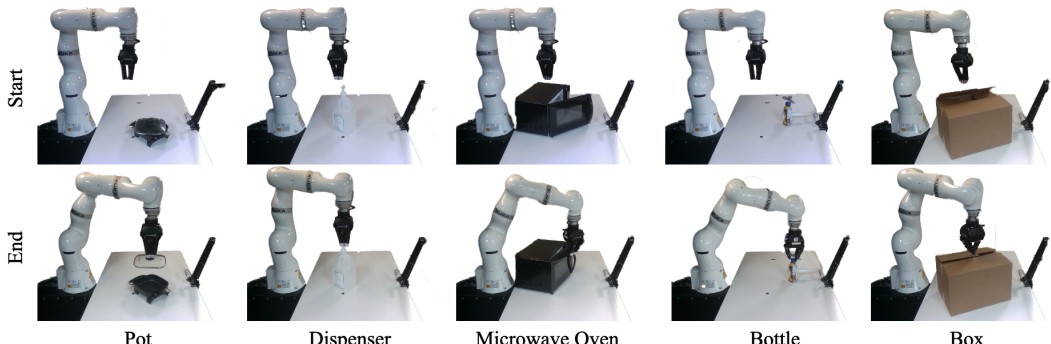

Figure 7: Real-world manipulation experiments using a Kuka robot with a Robotiq hand.

## 5 Conclusions

We present the Actionable Articulation-Aware Vision Language Model (A3VLM), an object-centric robot VLM designed to understand the articulation and action affordances of articulated objects. Unlike previous action-centric robot VLMs, A3VLM does not require any robot interaction data and can be adapted to various robot configurations. Although it is trained solely on simulated data, A3VLM demonstrates significant manipulation capabilities and inference stability across both simulation benchmarks and real-world robot experiments, establishing a new state of the art in this area. We provide further discussions on the limitations in the Appendix. We believe that A3VLM represents a promising direction for future research in robot VLMs.

**Acknowledgments**

This project is funded in part by the National Key R&D Program of China Project 2022ZD0161100, by the Centre for Perceptual and Interactive Intelligence (CPII) Ltd under the Innovation and Technology Commission (ITC)'s InnoHK, by Smart Traffic Fund PSRI/76/2311/PR, by RGC General Research Fund Project 14204021. Hongsheng Li is a PI of CPII under the InnoHK. Moreover, this work is partially supported by NSF awards 1846043 and 2132972.

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

## A  Limitation and Future Direction

Although A3VLM shows promising results in both simulation benchmarks and real-world experiments, several limitations remain. First, A3VLM currently operates in an open-loop manner, with no feedback process integrated into the current setup. Exploring how to incorporate feedback into this object-centric pipeline is a valuable direction for future research. Secondly, the z value in A3VLM's output is based on a normalized depth, which requires the user to provide a depth range for the scene to convert it back to real depth. Combining A3VLM with image-to-depth models could address this limitation. Lastly, A3VLM currently focuses on generating single-step manipulation for articulated objects. However, this object-centric representation can be extended to tasks such as pick and place, tool usage, and more. Therefore, expanding A3VLM's representation to encompass a broader range of manipulation tasks is another promising direction.

## B  Further Ablations

To study the effectiveness of the proposed methods, we conducted the following experiments:

- One-Step Reasoning: Instead of the multiple-round QA in the default A3VLM, we formulated the articulated object manipulation tasks in a one-step format, outputting the 3D bounding box and attached axis representation in a single step.
- Lower Image Resolution (224 pixels): We trained a model using images with a resolution of 224x224 pixels. This is done by discarding the sub-patches techniques used in our default setting.

As shown in Table 4, one-step reasoning resulted in a slight performance degradation compared to the default setting. We believe the chain-of-thought (CoT) approach reduces task reasoning difficulty. Additionally, results with lower image resolution align with our previous findings with LLaVA, highlighting that higher resolution aids in capturing fine-grained structures and improves generalization ability. Larger image resolutions (beyond 448 pixels) are not supported by the chosen MLLM due to limitations in the original visual foundation models, which generally accept images with 224 pixels.

Table 4: Ablation Studies.

| CoT Inference | Higher Img Res | Train-Category | Test-Category | All-Avg |
|:---:|:---:|:---:|:---:|:---:|
| ✓ | | 0.87 | 0.74 | 0.85 |
| ✓ | | 0.88 | 0.50 | 0.82 |
| ✓ | ✓ | 0.91 | 0.76 | 0.88 |

## C  Prompt Engineering for action affordance generation

## D  Real-world Experiments Details

Since A3VLM is trained on purely simulated data, it is necessary to address the discrepancies between simulation and real-world data. In simulations, the image resolution is 960x960, the focal length is 1000 pixels, and the rendered object images lack backgrounds. To align real-world images with those of the simulation, we first tuned the real-world camera's intrinsics to match those of the simulation using a homography transformation. Then, we used the Segment-Anything-Model (SAM) [34] to segment out the background.

It is worth noting that the output z-value of the 3D bounding box is not an absolute value but a normalized one. To denormalize it, we need to know the minimal and maximal depths of the scene. For this purpose, we use the depth images from the depth camera. Although real-world depth images

```
Role and Task Description:
 Develop a systematic approach for generating grounding tasks involving object links, where each task
 involves a limited number of steps and utilizes predefined action primitives from a robot skill library.

Robot Skill Library::
Actions available include:
"slide_open", "slide_close", "flap_open", "flap_close", "cap", "uncap", "pick", "place", "slide_in",
     ↪ "slide_out", "wipe", "press", "rotate", "StatusComplete".

Requirements and Constraints:

 1.  Tasks and actions must be tailored based on the current status of the link .
 2.  Links may have different joint types:  prismatic, revolute, static, etc.
 3.  All actions must be sourced exclusively from the provided skill library.
 4.  Provide the list of tasks and corresponding actions in JSON format.
 5.  The generated actions should vary in sequence length, order, and semantics.
 6.  Create tasks involving both single and multiple links where applicable.
 7.  Do not assume or add components not explicitly specified.

Examples:
   examples ...

   avilible links information ...

Instruction:
 Now please generate the tasks and actions for the OBJECT_CLASS's link part with the links LINK_INFO.
 You have generated tasks and actions in the previous as following HISTORY_GENERATION, please make sure the
 tasks and actions are different from the previous ones.

Please ONLY generate the tasks and actions in the valid json format.
```

Listing 1: An example prompt for guiding ChatGPT to generate the action grounding task for a specific object.

are full of noise and inaccuracies, we can still use the minimal and maximal depth values, as the relative error is small.

Regarding the robot experiments, the robot setup is illustrated in Fig. 8. We translate A3VLM's output into actions using the primitives described in Section 3.5. For generating grasp pose candidates, we can employ tools like the Grasp Pose Generator (GPG) [35], GraspNet [36], or define them manually. Since this is not the primary focus of our work, we have simplified the process by manually providing a list of grasp poses.

## E   Discussion on Input Modality

A3VLM takes RGB images as input. Compared with other methods that rely on depth images or point clouds, RGB suffers less noise in real-world experiments. However, as A3VLM is learning a 3D bounding box, a natural concern is whether pure RGB input is less accurate compared with depth input. To explore this problem, we trained a new version of A3VLM, i.e. A3VLM-depth, using depth images as input. To adapt the depth images for the A3VLM pipeline, we first normalized the depth values to the range [0, 1] and then converted them to RGB values.

We evaluated the performance of A3VLM and A3VLM-depth on the PartNet-Mobility simulation benchmark as described in Section 4.2. The performance is shown in Tab. 5. From the results, we can see that A3VLM and A3VLM-depth perform similarly on the training categories. However, in the testing categories, A3VLM shows a significant improvement, indicating that the pure RGB input is actually better for generalization. This ablation study confirms our initial hypothesis of using pure RGB as the input modality.

## F   Exploration on More Input Modalities

In addition to the modalities discussed in Section E, we also explored expanding our experiments to include RGB-D and point cloud modalities.

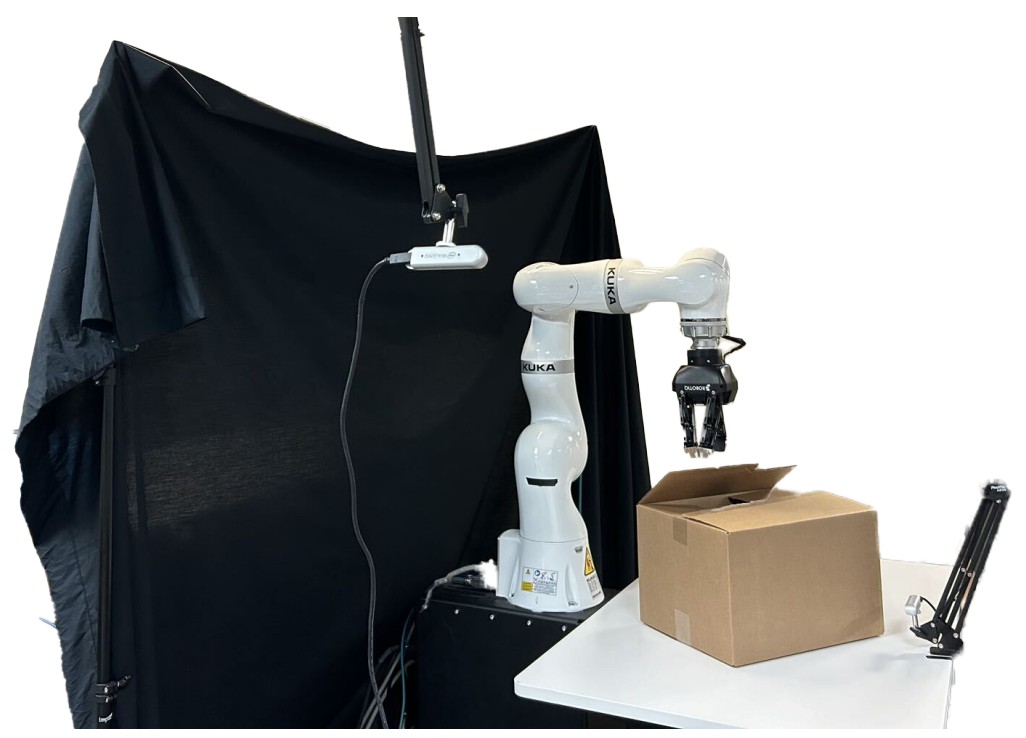

Figure 8: Robot setup: we use a Kuka robot with a three-fingers Robotiq hand. A realsense d415 camera is placed at the side to capture RGBD images. The depth camera on the table is not used for this experiment.

| Method | | | | | | | | | | | | | | | | |
|---|---|---|---|---|---|---|---|---|---|---|---|---|---|---|---|---|
| A3VLM | 0.90 | 0.82 | **0.94** | 0.90 | **0.49** | **0.70** | 0.87 | 0.35 | **0.86** | 0.79 | **1.00** | **0.70** | **0.83** | **0.97** | **0.34** | **0.40** |
| A3VLM-depth | **0.92** | **0.89** | 0.93 | **1.00** | 0.35 | 0.56 | **1.00** | **0.63** | 0.72 | **0.87** | **1.00** | 0.67 | 0.80 | 0.95 | 0.29 | 0.36 |

| Method | | | | | AVG | | | | | | | | | | | AVG |
|---|---|---|---|---|---|---|---|---|---|---|---|---|---|---|---|---|
| A3VLM | 0.62 | 0.50 | **0.90** | 0.73 | **0.91** | **0.88** | 0.76 | **0.74** | **0.86** | 0.79 | 0.67 | **0.96** | 0.50 | **0.62** | **0.72** | **0.76** |
| A3VLM-depth | **0.83** | **0.65** | 0.85 | **0.76** | 0.90 | 0.82 | **0.77** | 0.68 | 0.73 | **0.79** | 0.48 | 0.94 | **0.67** | 0.57 | 0.50 | 0.70 |

Table 5: Comparisons of A3VLM against A3VLM-depth. The first 20 categories are training categories and the following 10 categories are testing categories. A3VLM with RGB as the input modality has better performance and generalization ability.

## F.1 RGB-D Modality

For the RGB-D experiments, RGB and depth images were processed separately using distinct visual encoders. The outputs, termed visual tokens, were appended sequentially—with special tokens $$ and $<depth>$ indicating their respective modalities—and then fed into the Large Language Model (LLM). However, the LLM struggled with instruction following, likely due to two main challenges:

- Domain Gap: The visual foundation models, originally pre-trained only on natural RGB images, fail to reliably extract features from depth images that lack visual textures.

- Token Length: The combination of tokens from both modalities resulted in excessively long input sequences, which the LLM could not process effectively due to its limitations in handling long sequences.

### F.2  Point Cloud Modality

For point cloud inputs, we utilized PointBert [37] and RECON [38] as the point encoders, following practices established in ShapeLLM [38] and PointLLM [39]. The point features were aligned with language features using the Cap3D [40] dataset during fine-tuning of the projection layers. Despite successful training where the Modified Large Language Model (MLLM) produced high-quality captions, the LLM failed to predict partial object bounding box coordinates. This difficulty is attributed to:

- Lack of Visual Texture: Point clouds inherently contain fewer visual texture features, complicating the task of partial-level object detection.
- Model and Data Limitations: The point cloud models, having fewer parameters and a smaller training dataset compared to visual foundation models, exhibit weaker performance capabilities.

## G  Data Augmentation Examples

In this section, we displayed more data augmentation examples in Figure 9. The first row and third rows display the raw images rendered directly from the PartNet Mobility, while the second row and the last row display the generated ones. And we used the prompts in the List. 2 to guide the ChatGPT to generate more diverse texture descriptions for the target object.

```
Role and Task Description:
 You are a good assistant, skilled in providing accurate prompts for stable diffusion.

 I want to use stable diffusion to draw a category, please give me ten prompts with different styles.

 Note that the target object is the daily-use item.

 I already have the descriptions like previous_description, please avoid repeats.

 Only give me the newly generated prompts in a list, and nothing else.
```

Listing 2: An example prompt for guiding ChatGPT to generate diverse texture description for a specific object.

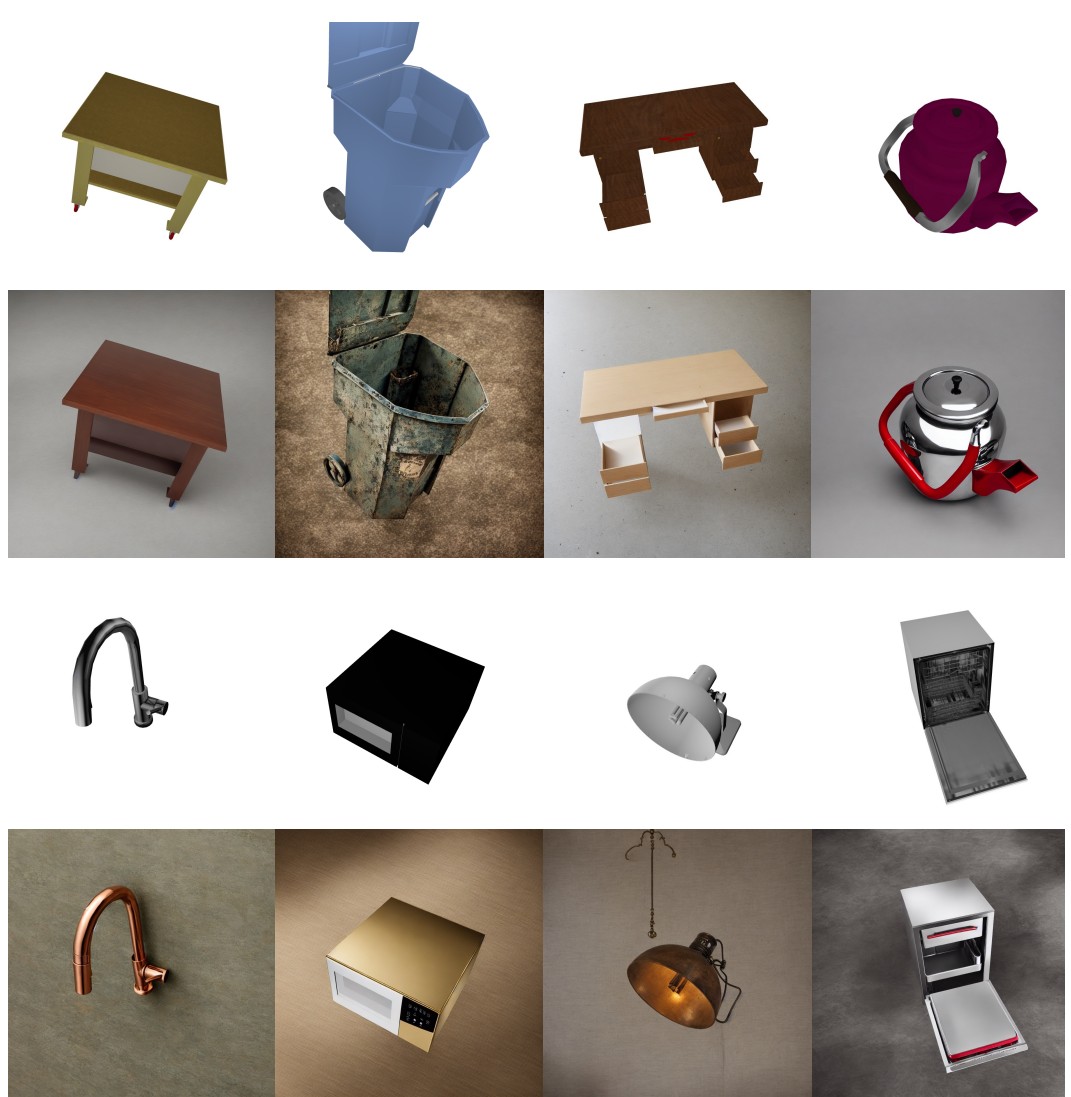

Figure 9: Comparison between raw images with the Stable Diffusion generated ones.

