# OpenReview forum: "A3VLM: Actionable Articulation-Aware Vision Language Model"
_robot-learning.org/CoRL/2024/Conference — CoRL 2024_

### Official Review · Reviewer_W221 · 2024-07-17
**A3VLM manuscript review**

**Originality:** 4
**Technical Quality:** 4
**Clarity Of Presentation:** 5
**Potential Impact:** 3
**Recommendation:** 3
**Confidence:** 5

**Review:**

The authors present a promising method for fine-tuning an open-source VLM model to interact with complex household objects, focusing on object-centricity and articulation awareness. While the approach demonstrates clear merit, the evaluation strategy could be strengthened to more rigorously demonstrate the model's effectiveness and maximize the impact of this work. Specific suggestions for improvement are detailed below:

**Strengths**:
- Robot-Agnostic Generalization: The object-centric representation learning approach employed by A3VLM facilitates knowledge transfer across diverse robotic platforms, requiring minimal platform-specific data and enhancing the model's potential for real-world deployment.
- Real-World Applicability: The explicit modeling of object articulation within A3VLM provides a robust framework for understanding and interacting with complex, articulated objects commonly encountered in real-world scenarios.
- The manuscript clearly and concisely communicates the main research contributions, methodology, and experimental results.

**Weaknesses**
- Evaluation Metric Limitations: The evaluation metric employed in the study, specifically the 1cm part movement success criterion, does not adequately reflect real-world scenarios. In practical applications, this criterion may not be indicative of meaningful object manipulation. Furthermore, the metric fails to account for potential damage caused by excessive force or torque applied orthogonal to the articulation axis, a critical consideration when interacting with articulated objects. Consequently, the results presented in Table 2 may not be fully representative of real-world performance.
- Simulation Experiment Concerns: The use of an arbitrarily small gripper model in certain simulation experiments (e.g., pliers, suitcase, laptop, toaster) raises concerns regarding the generalizability of the findings to real-world scenarios. Physical limitations and potential grasp-related issues in real-world manipulation may significantly impact performance, underscoring the need for caution in interpreting the simulation results.
- Real-World Experiment Limitations: The real-world experiments do not adequately demonstrate the potential consequences of inaccurate articulation parameter estimation. For instance, the pot lid pick-up experiment does not involve a meaningful articulation, limiting its relevance for assessing the VLM's ability to learn effective parameters. The inclusion of grasping and subsequent manipulation in real-world experiments would provide a more comprehensive evaluation of the VLM's capabilities.

**Minor updates**:
- Please add detailed captions in all figures and tables, that briefly describe the salient points of the plots and figures, explaining the axes, legends, and main takeaways.
- In Figure 4, please explain in the caption what different symbols mean, e.g., the fire emoji.
- Please prefer using the term `end-effector` instead of `manipulator` in the text. Usually, in the robotics literature, the term `manipulator` refers to the complete robot arm, and the term `end-effector` refers to the gripper attached at the end of the robot arm.
- Show the speed-up factor in the included videos for the experiments.
- Define `sufficient distance` in line 255
- Prefer citing thoroughly in the text, e.g., cite PartNet-Mobility in line 120, PyRender in line 122.

**Potential Future Extensions**:
Jain et. al. [1] proposed a category-independent articulation model parameterization to represent common articulation types. Using the proposed screw parametrization as the semantic label $\mathcal{S}$ should simplify the formulation while maintaining its generality.

**References**:
- [1] A. Jain, R. Lioutikov, C. Chuck and S. Niekum, "ScrewNet: Category-Independent Articulation Model Estimation From Depth Images Using Screw Theory," 2021 IEEE International Conference on Robotics and Automation (ICRA), Xi'an, China, 2021, pp. 13670-13677, doi: 10.1109/ICRA48506.2021.9561132.

**Quality Of The Limitations Section:**

2

**Questions For Rebuttal:**

Q1. Could the evaluation metric be refined to better account for potential damage caused by excessive force or torque applied orthogonal to the articulation axis, a critical consideration for real-world articulated object interaction? Given the limitations of the current metric, to what extent are the results presented in Table 2 truly representative of real-world performance?

Q2. How is the use of an arbitrarily small gripper model in certain simulation experiments (e.g., pliers, suitcase, laptop, toaster) justified?

**Robotics Focus:**

4

**Summary Of Paper:**

The authors in this work propose a vision language model (VLM), A3VLM, that is object-centric and articulation-aware. Prior work has focused on directly learning robot-centric data for interacting with objects, requiring them to collect a significant amount of robot interaction data that is costly in the real world. A3VLM, instead, focuses on the articulation structure and action affordances of objects, which helps it to be robot-agnostic. This approach is validated through experiments in both simulated benchmarks and real-world scenarios, demonstrating the effectiveness of A3VLM for a variety of object interaction tasks.

**Summary Of Recommendation:**

The authors present a promising method for fine-tuning an open-source VLM model to interact with complex household objects, focusing on object-centricity and articulation awareness. While the approach demonstrates clear merit, I have reservations about the evaluation strategy used to evaluate the proposed approach. If the authors can provide additional experimental analysis to address my concerns, I would be willing to accept the paper for publication.

---

### Official Review · Reviewer_FBJx · 2024-07-19
**The strength and weakness of A3VLM**

**Originality:** 4
**Technical Quality:** 4
**Clarity Of Presentation:** 4
**Potential Impact:** 3
**Recommendation:** 3
**Confidence:** 4

**Review:**

**Strengths**:
This paper provides an unified structure regardless of the robotic structure to predict the 3D bounding box of the target object and controll the end effector for mainpulation.

The authors design a novel chain-of-thought to build the framework by first predict the action type and bounding box for the movable part of the mentioned object and then predict the articulation structure of the object.

The analysis could be utilized to controll the robot for manipulation.

The data collection process is reasonable and the constructed dataset is valuable for the community.

The model performance in the simulator surpasses all the baseline methods by a large margin, and the realworld experiments are promising.
All the data and code are open-sourced.

**Weakness**:

The limitations of the paper is not well addressed.

The comparison baselines could involve some other vision-language models like llava, gpt4-o to generate the required coordination information in the same process.

More ablation studies like chaning the chain-of-thought or directly predict all the required spatial information, improve the resolution of the image should be added.

**Quality Of The Limitations Section:**

2

**Questions For Rebuttal:**

See weakness

**Robotics Focus:**

4

**Summary Of Paper:**

This paper provides an unified structure to predict the 3D bounding box of the target object and controll the end effector to manipulate. The fully data collection, training process and experiments demonstrates the effectiveness of the work.

**Summary Of Recommendation:**

This paper provides an unified structure regardless of the robotic structure to predict the 3D bounding box of the target object and controll the end effector for mainpulation. The authors design a novel chain-of-thought to build the framework by first predict the action type and bounding box for the movable part of the mentioned object and then predict the articulation structure of the object. The model performance in the simulator surpasses all the baseline methods by a large margin, and the realworld experiments are promising. Thus i recommand accept this paper.

---

### Official Review · Reviewer_tJSN · 2024-07-21
**Though it tackles a specific types of motions/articulation, it provides a practical method that leverages VLM effectively only from simulation but achieves good performance and that has a potential to be generalized to various robots.**

**Originality:** 3
**Technical Quality:** 4
**Clarity Of Presentation:** 4
**Potential Impact:** 3
**Recommendation:** 3
**Confidence:** 3

**Review:**

### Strengths
* Straight forward approach with strong performance on the benchmark.
* Does not require the real-world data collection.
### Weaknesses
* It follows the very same setup as ManipLLM did. If it is generalizable to different robots, it would be better to apply on another robot and compare the performance. It is hard to understand the contribution of the choice of the primitives.
* Limited to the specific actions. If it was effective to fine-tune VLM with carefully simulated data to annotate bounding boxes, it can be also applied to pick and place tasks which can make the model more general and comparable to various models.
* Normalized depth representation requires a depth bound. It is not sure if it can be inference-time generalizable to different depth ranges. This may make the approach less practical to be applied to tasks that are not bounded in a table-top setup. (It uses a depth image to determine the scene bound for depth normalization; it makes the method tightly coupled with the scene and potentially not generalizable to different scenes.)
* Not well explained about why depth or RGBD models did not work well or worse. Leveraging depth can boost the accuracy in 3D in general.

**Quality Of The Limitations Section:**

1

**Questions For Rebuttal:**

* Depth representation
  * Is the normalized depth best? I wonder if you have tried another representation for 3D bounding-box or depth representation.
  * Are the depth range values fixed during the test and the training? Are they the same?
  * I wonder if this is sensitive to depth boundaries.
* RGBD model
  * I wonder if you tried multi-modal models that also encodes and takes depth images in training.
  * Also, combined with SAM, a bounding-box prediction with an RGBD image might be more efficiently done without assuming the depth range.
* I guess it would be good to have some comparison to robot-centric approaches.
* It would be good to have some ablation on accuracy vs. success rate. (e.g. is an axis error affecting more than the bounding-box error? how tolerant motion primitives are w.r.t axis error or bounding-box error?)
* The model only predicts the bounding-box and axis but not the angle or progression. Was the experiment done in closed-loop control? How fast it is? How the action is done without knowing it is done? Is there any reason not training the progression or angle?

**Robotics Focus:**

4

**Summary Of Paper:**

It proposes a fine-tuned VLM to predict articulation parameters from an image and demonstrates that it can be performant combined with action primitives.

**Summary Of Recommendation:**

Recommend weak accept. It was easy to read, it proposes a straight-forward approach with good result from a fair comparison. It can be useful in practice if we can combine different tasks in fine-tuning.

---

### Author Rebuttal · Authors · 2024-08-13

We are grateful to all the reviewers for their valuable feedback. We are encouraged by the positive reviews, such as: "it proposes a **straightforward approach with good results** from a fair comparison" (Reviewer tJSN), "The authors design a **novel chain-of-thought** to build the framework" (Reviewer FBJx), and "The authors present a **promising method** for fine-tuning an open-source VLM model to interact with complex household objects."(Reviewer W221).

In response to your feedback, we have made the following improvements:

1. **Depth Modality and Bounding Box Representation**: We addressed the minor issues on depth modality usage and bounding box representation pointed out by Reviewer tJSN.
2. **Baselines, Limitations and Ablation Studies**: We added more baselines, including LLaVA and GPT4o, and conducted ablation studies on image resolution and CoT inference, following Reviewer FBJx's advice. And we further updated the limitation part in our revised paper.
3. **Evaluation Metric and Success Threshold**: We included detailed discussions on the evaluation metric and conducted experiments on success threshold settings based on Reviewer W2211's advice.
4. **Detailed Responses**: While the core content of our paper remains largely unchanged, we have provided detailed responses to the questions and concerns raised by all reviewers. These responses offer additional context, clarifications, and explanations.

We have attached a revised PDF incorporating the minor corrections and updated real-world demos with a visualized speed-up factor. Our detailed responses in the comments section provide comprehensive answers to the specific questions and concerns raised by each reviewer.

We believe these revisions and our detailed responses effectively address the feedback and highlight the significance of our work. Once again, we thank the reviewers for their valuable input and are confident that these clarifications and improvements enhance the overall quality and clarity of our paper. Our group is willing to reply to more questions raised by the reviewer during the rest of rebuttal time.

---

### Decision · Program_Chairs · 2024-09-04

**Decision:**

Accept

**Comment:**

The paper is well motivated, timely, tackles a relevant problem for the robotics community, contains sufficient novelty, is well written, and sufficient experiments are provided.

The requests of the reviewers during the rebuttal phase have sufficiently been addressed.

I recommend acceptance.